# Adsorption of REEs from Aqueous Solution by EDTA-Chitosan Modified with Zeolite Imidazole Framework (ZIF-8)

**DOI:** 10.3390/ijms22073447

**Published:** 2021-03-26

**Authors:** Sihan Feng, Xiaoyu Du, Munkhpurev Bat-Amgalan, Haixin Zhang, Naoto Miyamoto, Naoki Kano

**Affiliations:** 1Graduate School of Science and Technology, Niigata University, 8050 Ikarashi 2-Nocho, Nishi-ku, Niigata 950-2181, Japan; f19b138f@mail.cc.niigata-u.ac.jp (S.F.); duxiaoyu90@163.com (X.D.); munkhpurev@must.edu.mn (M.B.-A.); F19B137H@mail.cc.niigata-u.ac.jp (H.Z.); 2Department of Chemistry and Chemical Engineering, Faculty of Engineering, Niigata University, 8050 Ikarashi 2-Nocho, Nishi-ku, Niigata 950-2181, Japan; nmiyamoto@eng.niigata-u.ac.jp

**Keywords:** ethylenediamine tetra-acetic acid–chitosan (EDTA–CS), zeolite imidazole framework (ZIF-8), rare-earth elements (REEs), adsorption capacity, partition coefficient

## Abstract

Chitosan (CS) modified with ethylenediamine tetraacetic acid (EDTA) was further modified with the zeolite imidazole framework (ZIF-8) by in situ growth method and was employed as adsorbent for the removal of rare-earth elements (REEs). The material (EDTA–CS@ZIF-8) and ZIF-8 and CS were characterized by X-ray diffraction (XRD), Fourier transform infrared spectroscopy (FT-IR), scanning electron microscope (SEM), and nitrogen adsorption/desorption experiments (N_2_- Brunauer–Emmet–Teller (BET)). The effects of adsorbent dosage, temperature, the pH of the aqueous solution, contact time on the adsorption of REEs (La(III), Eu(III), and Yb(III)) by EDTA–CS@ZIF-8 were studied. Typical adsorption isotherms (Langmuir, Freundlich, and Dubinin–Radushkevich (D-R)) were determined for the adsorption process, and the maximal adsorption capacity was estimated as 256.4 mg g^−1^ for La(III), 270.3 mg g^−1^ for Eu(III), and 294.1 mg g^−1^ for Yb(III). The adsorption kinetics results were consistent with the pseudo-second-order equation, indicating that the adsorption process was mainly chemical adsorption. The influence of competing ions on REE adsorption was also investigated. After multiple cycles of adsorption/desorption behavior, EDTA–CS@ZIF-8 still maintained high adsorption capacity for REEs. As a result, EDTA–CS@ZIF-8 possessed good adsorption properties such as stability and reusability, which have potential application in wastewater treatment.

## 1. Introduction

In recent years, water pollution has attracted great attention. At present, heavy metal pollution is one of the most serious environmental problems and has received great global attention by experts in various fields. If wastewater containing toxic heavy metals is directly discharged into the environment without being treated, it poses a great threat to ecology and human health. In addition, metal elements including rare-earth elements (REEs) have broad industrial-application prospects, but with the discharge of wastewater into the environment, they hugely impact it. From the point of view of water-resource utilization and environmental protection, it is essential to explore an efficient environmental protection method for rare-earth wastewater treatment [1]. The early stage of the rare-earth industry chain mainly includes rare-earth mineral processing, development, mining and smelting separation, and rare-earth wastewater is primarily produced in the rare-earth smelting process. If management is improper, rare-earth mining, smelting, and separation seriously impact the environment. Therefore, these rare-earth elements need to be removed as much as possible before discharging them into the underground environment. Currently known methods for separating and recovering REEs from aqueous solutions include extraction [2,3], adsorption [4,5,6], and chemical precipitation [7]. Among them, most methods are costly and require much energy consumption, are harmful to the environment, and are inefficient in processing low concentrations of rare earth elements [8]. Considering technology, the economy and environmental protection, the adsorption method was chosen as the method to remove rare-earth elements from pollutants [9].

Among methods for removing heavy metals from wastewater, biosorption has recently attracted more research interest [10,11,12]. Adsorption is a method to remove water pollution by the adsorption and complexation of adsorbents. Because of its high efficiency, low cost, low energy consumption, and reusability, it is widely used in the treatment of heavy metal pollution in water. Activated carbon, inorganic minerals, polymers, nanoparticles, and other materials can be used as adsorbents [13,14,15,16]. In the past few decades, researchers from various countries studied various biomass materials in order to achieve the goal of using low-cost biological materials as adsorbents for heavy metals in wastewater. Among available biosorbents, chitosan (CS) is easy to obtain, and because it is positively charged, and the active functional groups in the molecule can be cross-linked, grafted, and alkylated with other reagents, it has strong adsorption capacity [17,18,19,20,21]. With the rapid development of new materials, metal organic frameworks (MOFs) have received more attention in recent years [22]. MOFs are crystalline materials with a porous network structure. They have the advantages of low density, large specific surface area, wide variety, structural design, and controllability. Therefore, MOFs are widely used in adsorption separation [23], sensors [24], photocatalysis [25], and drug delivery [26]. Among them, the study of metal organic frameworks as adsorbent to remove heavy metals in water has become the focus of attention [27]. In fact, MOFs have been proved to be good adsorbents for substances including heavy metals. Zeolitic imidazolate frameworks (ZIFs) are a type of MOF material with a zeolite framework structure. This type of material combines the high stability of zeolite and the diverse structure and performance of MOFs. ZIF-8 as the most representative ZIFs material has been widely concerned. The synthesis method of ZIF-8 is simple and the yield is high [28,29], but ZIF-8 still has some shortcomings, such as easy aggregation due to the characteristics of nanoparticles and easy loss during recycling.

In this experiment, chitosan was modified with ZIF-8 by an in situ growth method. According to reports, zeolitic imidazolate framework-8 can effectively adsorb and remove Cr(VI) ions in aqueous solutions [30]. In addition, there are studies on preparing chitosan/ZIF-8 composite beads to efficiently remove U(VI) [31]. However, few studies used ZIF-8-modified chitosan as an adsorbent for rare-earth elements. Considering the above discussion, the combination of ZIF-8 and chitosan may cause chitosan to lose some adsorption sites, resulting in a decrease in adsorption capacity, so this experiment also introduces ethylenediamine tetraacetic acid (EDTA) to compensate for some of the lost adsorption sites. EDTA is a common cross-linking and chelating agent that is widely used in industry and agriculture. It was grafted onto chitosan molecular chains to increase the adsorption points of chitosan molecules, thereby increasing the adsorption capacity of ions. Thus, in this study, we synthesized a new type of adsorption material EDTA–CS@ZIF-8. In order to evaluate new adsorption material EDTA–CS@ZIF-8, the X-ray diffraction (XRD), Fourier transform infrared spectroscopy (FT-IR), scanning electron microscopy (SEM), and Brunaeur–Emmet–Teller (BET) methods were used to determine the crystal types, functional groups, surface morphology, and specific surface area of the materials. In this work, La(III), Eu(III), and Yb(III) were used to carry out adsorption experiments using EDTA–CS@ZIF-8 to adsorb REEs from aqueous solutions. The effects of adsorbent dosage, temperature, pH, contact time, and competing ions on the removal of REEs from aqueous solutions were studied. After obtaining the optimal conditions for the adsorption of rare-earth ions, the adsorption capacity of REEs on EDTA–CS@ZIF-8 was discussed. In order to apply adsorbents to various fields, adsorption kinetics and adsorption isotherm models were established to interpret the data and determine the extent of the adsorption process [32,33]. The purpose of this research is to synthesize a new type of adsorption material EDTA–CS@ZIF-8, and to investigate its adsorption efficiency for rare-earth elements (La(III), Eu(III), and Yb(III)) in water, aiming to develop a method for the efficient adsorption and recovery of heavy metals and to apply it in practice. The results of this research may also be used for the recovery and reuse of rare metal resources to provide help for the stable supply of rare metal resources.

## 2. Results and Discussion

### 2.1. Material Characterization

#### 2.1.1. XRD Patterns

The X-ray diffraction characterization results of EDTA–CS@ZIF-8, ZIF-8, and chitosan are shown in Figure 1. The CS curve showed wide peaks at 10° and 20°, which indicated that chitosan is amorphous. EDTA–CS@ZIF-8 showed a broad peak at 20°, which belonged to chitosan, indicating that EDTA–CS@ZIF-8 still retained the structure of chitosan. EDTA–CS@ZIF-8 showed characteristic peaks consistent with ZIF-8, proving that ZIF-8 was successfully cross-linked with chitosan.

#### 2.1.2. FT-IR Spectra

The FT-IR spectra of ZIF-8, chitosan and EDTA–CS@ZIF-8 are shown in Figure 2. The corresponding functional groups could be determined by analyzing the absorption peaks of the spectra. Chitosan had absorption peaks of –NH_2_ and –OH at 3600 cm^−1^, –CH stretching vibration at 3000 cm^−1^. The characteristic peaks of ZIF-8 from 600 to 1500 cm^−1^ correspond to the vibration of the imidazole ring, and characteristic peaks of ZIF-8 and CS could be found in EDTA–CS@ZIF-8. ZIF-8 cross-linked with chitosan. The spectrum showed that EDTA–CS@ZIF-8 had –NH_2_ vibrations at 1670 and 1596 cm^−1^. At the same time, the –NH absorption peak of EDTA–CS@ZIF-8 at 3600 cm^−1^ disappeared, and a new characteristic peak of C=O appeared at 1800 cm^−1^. These phenomena indicated that the acylation reaction occurred between EDTA and chitosan.

#### 2.1.3. SEM Micrographs

SEM images showed the surface features and interface interactions of pure CS beads and EDTA–CS@ZIF-8 beads. As shown in Figure 3a, the surface of pure CS beads was smooth and uniform. The image of the synthesized ZIF-8 is shown in Figure 3b. Figure 3c shows the further-enlarged images of pure chitosan beads. The surface of the enlarged chitosan beads was slightly rough. An SEM image of EDTA–CS@ZIF-8 beads is shown in Figure 3d. The surface of ZIF-8 became rough due to the growth of ZIF-8 on the surface of chitosan.

#### 2.1.4. BET Surface Area, Pore-Size, and Pore-Volume Analysis

The specific surface areas of ZIF-8, chitosan, and modified material EDTA–CS@ZIF-8 were determined by N_2_ adsorption. Analysis results of adsorption/desorption isotherms are shown in Table 1, which shows that the surface area and pore volume of EDTA–CS@ZIF-8 were decreased compared with those of ZIF-8, indicating that ZIF-8 successfully modified EDTA–chitosan. Compared with pure CS beads, the surface area and pore volume of EDTA–CS@ZIF-8 were increased, which is consistent with the SEM results, which provided the possibility of EDTA–CS@ZIF-8 composite beads to effectively adsorb REEs.

### 2.2. Evaluation of REE Adsorption Performance

#### 2.2.1. Effect of Adsorbent Dosage

Adsorbent dosage is an important factor affecting adsorption capacity. In order to determine the effect of adsorbent dosage on the adsorption of REEs, adsorbent dosage was varied (0.50–2.50 mg/L) under other fixed conditions (temperature: 298 K, pH: 7, contact time: 1 h, initial concentration: 1 mg/L). The results are shown in Figure 4. With the increase in dosage, adsorption capacity was obviously improved. For the dosage of 1.5 mg/L, more than 80% of the REEs were adsorbed, but when the dosage was greater than 1.5 mg/L, the adsorption amount did not significantly increase. Therefore, 1.5 mg/L was considered to be the optimal dosage for REE adsorption for the rest of the study.

#### 2.2.2. Effect of Temperature

With temperature as the only variable, the effect of REE adsorption by EDTA–CS@ZIF-8 was investigated. The study was carried out by varying the temperature (298–328 K) and keeping all other parameters unvaried (adsorbent dosage: 1.5 mg/L, pH: 7, contact time: 1 h, initial concentration: 1 mg/L). The results are shown in Figure 5. Temperature did not have an obvious effect on the adsorption of REEs, and the adsorption rate of REEs reached 80% at all temperatures levels. Then, 298 K, which is near room temperature, was chosen for the rest of this study from the viewpoint of environmental protection or energy conservation.

#### 2.2.3. Effect of Initial pH

The surface charge and metal complexation sites of the adsorbent, the ionization degree, and the form of the metal ions may have been affected by the pH value of the aqueous solution [34,35]. Generally, under low-pH conditions, hydrogen ions compete with rare earth cations for binding sites, making it difficult for rare-earth elements to be adsorbed. Under the condition of basic pH, negatively charged adsorption sites can fully attract positively charged REE ions. On the other hand, under high-pH conditions, REE ions form hydroxide precipitates [36]. Therefore, in this experiment, the effect of pH on REE was studied in the pH range of 4–7 (under the condition of adsorbent dosage: 1.5 mg/L, contact time: 1 h, initial concentration: 1 mg/L, temperature: 298 K). The results are shown in Figure 6. At this concentration, the pH range should be kept below 7.0 to prevent REEs from precipitating out of the solution in the form of hydroxide. The adsorption capacity of REEs increased with the increase in the pH value of the aqueous solution, and the adsorption capacity of REEs reached the maximum at pH 6. Therefore, pH 6 was chosen.

#### 2.2.4. Effect of Adsorption Time

The influence of contact time on the adsorption of REEs by EDTA–CS@ZIF-8 was explored. In this experiment, 15 mg of the adsorbent was used to adsorb the REE solution with an initial concentration of 1 mg/L at a temperature of 298 K. The pH value of the solution was maintained at 6 to achieve the best adsorption effect on REEs. The results from 1 to 48 h contact time are shown in Figure 7. This study was also be used to verify the adsorption kinetics of the adsorption process, which is discussed later in this work. The adsorption capacity of EDTA–CS@ZIF-8 for REEs increased sharply in the first hour, and until the contact time reached 24 h, the adsorption capacity of the adsorbent gradually increased. Approximately 90% of the REEs were removed from the solution within 24 h of contact time, and there was no significant change thereafter. Therefore, 24 h was chosen as the best contact time for further study.

Through the study of the adsorption process under different influencing factors, it was determined that EDTA–CS@ZIF-8 had the best adsorption effect on REEs under the following conditions: adsorbent dosage of 1.5 mg/L, pH of 6, the contact time of 24 h, and temperature of 298 K although the temperature had no obvious influencing effect on the adsorption of REEs.

#### 2.2.5. Effect of Competitive Ions

The effect of competitive ions on REE adsorption is shown in Figure 8. In this study, adsorption experiments of REEs were conducted under the presence of other competitive ions with different concentrations (i.e., 0, 25, 50, 100, and 200 ppm) of Na^+^, K^+^, Ca^2+^, and Mg^2+^. The initial concentration of REEs was 1 mg/L, pH was 6, contact time was 1 h, adsorbent dosage was 1.5 mg/L, and temperature was 298 K. The diagram shows that the adsorption capacity of EDTA–CS@ZIF-8 for La(III) slightly decreased as the concentration of competing ions increased, while it had little effect on Eu(III) and Yb(III). This shows that even in the presence of a high concentration of competing ions, EDTA–CS@ZIF-8 also showed good adsorption capacity for REEs and could be used as a rare-earth adsorbent.

### 2.3. Adsorption Kinetics Study

The kinetic model can be used to determine the mechanism of the adsorption process and provide effective data support for the study of the feasibility of process scaling [37,38]. In this study, the mechanism of the adsorption process was studied by fitting the experimental data to the pseudo first- and second-order reaction equations.

The pseudo first-order model is given by the following Equation (1):(1)$$\ln\left(q_{e}-q_{t}\right)=\ln\left(q_{e}\right)-k_{1}t$$
where *q_e_* and *q_t_* are the adsorption capacity (mg g^−1^) of REEs at equilibrium and time *t*, respectively, and *k*_1_ is the rate constant (h^−1^) of pseudo first-order adsorption.

The pseudo second-order rate equation is expressed as follows in Equation (2):(2)$$\frac{t}{q_{t}}=\frac{1}{kq_{e}^{2}}+\frac{t}{q_{e}}$$
where *k* is the rate constant of pseudo-secondary adsorption (g mg^−1^ h^−1^), and *q_e_* and *q_t_* are the adsorption capacity (mg g^−1^) of REEs adsorbed by the adsorbent at equilibrium and time t.

In order to examine the consistency between the model and the experimental results, under the optimized experimental conditions, pseudo-first and -second-order were used to apply the linear graphs of ln(*q*_e_ − *q*_t_) − *t* and *t*/*q*_t_ − *t* to the EDTA–CS@ZIF-8 REE adsorption kinetics model, as shown in Figure 9 and Figure 10. Table 2 shows the linear constants (*R*^2^) and other parameters of the two kinetic models for the EDTA–CS@ZIF-8 adsorption of REEs calculated according to Figure 9 and Figure 10. For the three REEs adsorbed by EDTA–CS@ZIF-8, their pseudo-second-order model had a higher *R*^2^ value than that of pseudo-first-order model. The higher *R*^2^ value indicated that the adsorption of REEs by EDTA–CS@ZIF-8 was more in line with the pseudo-second-order model, indicating that the adsorption process was mainly chemical adsorption. At the same time, the equilibrium adsorption capacity, calculated by fitting a straight line, was consistent with the experimental equilibrium adsorption capacity.

### 2.4. Adsorption Isotherm Study

In the process of adsorption, the study of adsorption isotherms is necessary and crucial to predict the adsorption behavior of pollutants onto the adsorbent surface. Typical adsorption isotherm (Langmuir, Freundlich, and Dubinin–Radushkevich (D−R)) models were used to evaluate the adsorption data.

The Langmuir isotherm model assumes that the surface of the adsorbent is uniform. Each surface molecule or atom of the adsorbent adsorbs the gas molecule, and the gas molecules adsorb on the solid surface as a single layer, and there is no force between the gas molecules adsorbed on the solid surface. It is given by the following Equation (3):(3)$$\frac{C_{e}}{q_{e}}=\frac{C_{e}}{q_{max}}+\frac{1}{K_{L}q_{max}},$$
where *C_e_* and *q_e_* are the REE concentration (mg L^−1^) and adsorption capacity (mg g^−1^) when adsorption reaches equilibrium, *q_max_* is the maximal adsorption capacity of the adsorbent (mg g^−1^), and *K_L_* is the adsorption constant of Langmuir isotherm (L mg^−1^). The relationship between *C_e_*/*q_e_* and *C_e_* gives a straight line with slope of 1/*q_max_* and intercept of 1/(*K_L_q_max_*).

The Freundlich isotherm model is a multilayer adsorption process without considering the adsorption saturation, which occurs on the multi-layer heterogeneous surface. The isotherm of the linear Freundlich model is represented by Equation (4):(4)$$\ln q_{e}=\ln K_{F}+\left(1/n\right)\ln C_{e},$$
where *K_F_* is adsorption capacity ((mg g^−1^) (dm^−3^ mg^−1^)^1/*n*^), and 1/*n* is adsorption strength. The relationship between ln*q_e_* and ln*C_e_* determined the 1/*n* and *K_F_* values. The value of 1/*n* could be used to judge the difficulty of adsorption process: irreversible adsorption (1/*n* = 0) favorable adsorption (0 < 1/*n* < 1) or unfavorable adsorption (1/*n* > 1) [39].

The Langmuir–Freundlich isotherm is a three-parameter isotherm, which describes the distribution of adsorption energy on the heterogeneous surface of the adsorbent [32]. At low adsorbate concentration, the model becomes the Freundlich isotherm model, and at high adsorbate concentration, the model becomes the Langmuir isotherm. The Langmuir–Freundlich isotherm can be expressed by Equation (5):(5)$$q_{e}=\frac{q_{MLF}{\left(K_{LF}C_{e}\right)}^{M_{LF}}}{1+{\left(K_{LF}C_{e}\right)}^{M_{LF}}}$$

Among them, *q_MLF_* is the maximum adsorption capacity of Langmuir–Freundlich (mg g^−1^), *K_LF_* is the equilibrium constant of heterogeneous solids, and *M_LF_* is the heterogeneous parameter, between 0 and 1.

The adsorption process is also analyzed by the Dubinin–Radushkevich (D−R) isotherm model to confirm whether the adsorption process is chemical adsorption or physical adsorption. The D−R isotherm model is valid in the low concentration range and can be used to describe adsorption on homogeneous and heterogeneous surfaces. The mathematical equation of the D−R isotherm adsorption model can be expressed as Equation (6):(6)$$\ln q_{e}=lnq_{0}-K\varepsilon^{2}$$
where *q_e_* is the adsorption capacity of the REEs when the adsorption reaches equilibrium (mol g^−1^); *q_0_* is the theoretical saturation capacity (mg**·**g^−1^); *K* is the activity coefficient related to the average free energy of adsorption (mol^2^**·**kJ^−2^); *ɛ* is the Polanyi potential (kJ**·**mol^−1^), which can be calculated by Equation (7):(7)$$\varepsilon =\mathrm{RTln}(1+1/C_{e}).$$

Among them, *R* is the universal gas constant (8.314 J**·**mol^−1^**·**K^−1^); *T* is the Kelvin temperature (K); *C_e_* is the concentration of the REEs when the adsorption reaches equilibrium (mol**·**L^−1^).

The D−R isotherm model is mainly used to estimate the average free energy of adsorption (kJ**·**mol^−1^), which can be calculated by Equation (8):(8)$$E=\frac{1}{\sqrt{2K}}.$$

The value of *E* can be used to distinguish the type of adsorption reaction: when 1 < E < 8 kJ**·**mol^−1^, adsorption is physical adsorption; when 8 < *E* < 16 kJ**·**mol^−1^, adsorption is dominated by ion exchange, and when it is greater than 20 kJ**·**mol^−1^, it is chemical adsorption [40,41].

Under optimal adsorption conditions (pH 6, 298 K temperature, 24 h contact time, and 1.5 mg·L^−1^ adsorbent dosage), the adsorption isotherm model of EDTA–CS@ZIF-8 for REEs was established at the initial concentration of 5–45 mg·L^−1^. The results of adsorption data investigated by Langmuir, Freundlich, and D−R equations are shown in Figure 11, Figure 12, and Figure 13, respectively. Other parameters including correlation coefficient (*R*^2^) are given in Table 3. From Table 3, each *R*^2^ value for REEs on EDTA-CS@ZIF-8 is comparatively large, and favorable adsorption of REEs by EDTA-CS@ZIF-8 was presented. Particularly, the adsorption was best fitted to the Langmuir adsorption isotherm equation, which suggests that the adsorption mainly occurred at the active sites on the surface of EDTA-CS@ZIF-8, that is, which mainly occurred by monolayer reaction. As mentioned above, Langmuir–Freundlich isotherms is derivations of Langmuir and Freundlich isotherms, and also supposed to be satisfactory fitted to the data because of high correlation coefficients in both Langmuir and Freundlich isotherms. From the Langmuir model, the maximal adsorption capacity of REEs were calculated to be 256.4 mg g^−1^ for La(III), 270.3 mg g^−1^ for Eu(III), and 294.1 mg g^−1^ for Yb(III), respectively. On the other hand, the correlation coefficients of the D-R isotherm model were above 0.98, indicating that the D-R isotherm model based on the theory of micropore adsorption could well describe the adsorption of EDTA-CS@ZIF−8 on REE in solution. According to the D-R isotherm adsorption model, the average free energy of the adsorption in this work was estimated to 12–14 kJ·mol^−^^1^, which showed that the adsorbing process of REE by EDTA-CS@ZIF-8 was chemical adsorption based on ion exchange.

### 2.5. Comparison of Maximal Adsorption Capacity and Partition Coefficient with Other Adsorbents in Previous Studies

Adsorption performance is been usually evaluated and expressed by maximal (or equilibrium) adsorption capacity in many adsorption studies. However, maximal adsorption capacity is sensitively influenced by the initial concentration of target pollutant. If the adsorbent is exposed to a higher adsorbate concentration, it is easy to show higher adsorption capacity. Similarly, if the adsorbent is exposed to a lower adsorbate level, adsorption capacity decrease. Therefore, in addition to the maximal adsorption capacity, it is effective to estimate it using the concept of partition coefficient (PC) [42,43,44]. Then, PC (Equation (9)) was also introduced in this work to evaluate adsorption performance:PC = adsorption capacity/final concentration.(9)

The comparison of the adsorption performance of REEs on various adsorbents in previous studies is listed in Table 4. The maximal adsorption capacity and PC of EDTA–CS@ZIF-8 were much larger than those of other adsorbents.

### 2.6. EDTA–CS@ZIF-8 Recyclability

The effective reuse of adsorbent materials can reduce the production and use costs of adsorbents. Therefore, in practical applications, the estimation of adsorbent stability and reusability of adsorbents is very important. In this study, a NaHCO_3_ (0.1 M) solution was used as the desorption agent, and desorption time was 1 h. EDTA–CS@ZIF-8 obtained through 5 adsorption/desorption cycles was investigated. Experimental conditions were as follows: the initial concentration of REEs was 1 mg·L^−1^, pH was 6, contact time was 3 h, adsorbent dosage was 1.5 mg·L^−1^, and the temperature was 298 K. The results of REE adsorption on the regenerated EDTA–CS@ZIF-8 are shown in Figure 14. For La(III), the adsorption capacity of EDTA–CS@ZIF-8 remained basically unchanged in the first three cycles, showing good repeated use performance. For Eu(III), the adsorption capacity of EDTA–CS@ZIF-8 decreased slightly after the first cycle, but adsorption capacity did not change in the subsequent cycles. For Yb(III), the adsorption capacity of EDTA–CS@ZIF-8 did not change much in the first 2 cycles. After 5 cycles, the adsorption capacity of EDTA-CS@ZIF-8 for La(III) was reduced to 62%, and adsorption capacity for Eu(III) and Yb(III) was reduced to 71%, but overall EDTA-CS@ZIF-8 still had a high capacity to adsorb REEs.

### 2.7. Adsorption Mechanism

Many adsorbent and adsorbate parameters, such as hydrophilicity and hydrophobicity, hydrogen and electrostatic repulsion and attraction, surface charge, functional groups (amine, hydroxyl, carboxyl, etc.), porosity and pore size, and the specific surface area of adsorbent affect the adsorption kinetics and the amount of removed from aqueous solutions. According to the literature [31,54,55], chitosan contains amine (–NH_2_) and hydroxyl (–OH), which can be used as coordination sites to form complexes with various heavy metal ions, so it can be used as an excellent biosorbent for heavy metals. In this study, the prepared adsorbent was EDTA–chitosan microspheres containing ZIF-8 on the surface. ZIF-8 has a large specific surface area, which increases the number of exposed active sites and improves the adsorption capacity. Some characterization methods were used in this study to determine the potential removal mechanism of REEs by comparing before and after the adsorption of Eu(III). As shown in Figure 15, the XRD patterns of EDTA–CS@ZIF-8 before and after the adsorption of Eu(III) were similar, indicating that there was no phase change after adsorption. From FT-IR spectra in Figure 16, it is found that the vibration of –NH_2_ at 1670 cm−^1^ almost disappeared after Eu(III) adsorption, which is the largest difference between the samples with lanthanides and empty, and that the broad peaks at 1596 and 3185 cm^−1^ are attributed to the stretching of –NH_2_ and –OH vibration. This indicates that there are a large number of –OH and –NH_2_ bonds on the surface of EDTA–CS@ZIF-8. These functional groups may have participated in the interaction with Eu(III) and have brought to the chelation of Eu(III) ions with imidazole and chitosan. In order to clarify the adsorption mechanism of EDTA–CS@ZIF-8 in more detail, XPS analysis was also performed, and results are shown in Figure 17. As shown in Figure 17a, the scanned XPS spectra of EDTA–CS@ZIF-8 showed C1s, N1s, O1s, and Zn2p3 peaks at 286.09, 400.05, 532.97, and 1022.31 eV, respectively. The O1s peak could be divided into four parts, of which 285.82 and 288.26 eV seemed to be the C–O and C=O bonds, respectively. The N1s peak could be divided into two peaks at 399.35 and 400.88 eV, which may have been –C–NH– obtained by the combination of EDTA and chitosan. The O1s spectrum showed peaks at 532.49 eV, corresponding to the peak of –OH. From Table 5, the main elements of EDTA–CS@ZIF-8 were carbon and nitrogen, and the atomic value of Zn element was 5.75%, which indicated that the synthesis of EDTA–CS@ZIF-8 was successful. Figure 17b shows the scanned XPS spectrum of EDTA–CS@ZIF-8 after adsorption of Eu(III). Compared with the spectrum before adsorption, the peak of Eu3d5 at 1137.45 eV has appeared after adsorption. From the separation peaks of N1s and O1s, the atomic values of N1s and O1s changed before and after adsorption. It suggests that Eu(III) reacted with nitrogen and oxygen on the surface of EDTA-CS@ZIF-8. Consequently, XPS analysis was consistent with the results of the above FT-IR experiments.

## 3. Materials and Methods

Chemical reagents, namely, methanol, zinc nitrate hexahydrate, 2-methylimidazole, chitosan, and acetic acid were purchased from Kanto Chemical Co., Inc. (Tokyo, Japan). REE nitrate salts such as La(NO_3_)_3_·6H_2_O, Eu(NO_3_)_3_·6H_2_O, and Yb(NO_3_)_3_·3H_2_O were purchased from Kishida Chemical Co., Inc. (Osaka, Japan). The La(III), Eu(III), or Yb(III) ion stock solution was prepared and suitably diluted with ultrapure water. All used reagents used were of analytical grade. During the whole working process, water (>18.2 MΩ) treated by the ultrapure water system (RFU 424TA, Advantech Aquarius, Suite A Dublin, CA, USA) was employed. pH value was measured with the pH meter (HORIBA F-72, Tokyo, Japan).

### 3.1. Adsorbent Synthesis

First, 2 g of chitosan and 4 g of EDTA were dissolved in 200 mL of an acetic acid solution (2.0%, *v*/*v*) and stirred for 48 h. Then, this was added dropwise to 0.5 M NaOH to obtain EDTA–chitosan microspheres. Subsequently, 3 g of zinc nitrate hexahydrate was dissolved in 100 mL of methanol solution and stirred at room temperature for 1 h. After 20 min, microspheres were taken out, washed with ultrapure water to neutrality, and immersed in a methanol solution of zinc nitrate hexahydrate. In addition, 6.6 g of 2-methylimidazole was dissolved in 100 mL of a methanol solution and stirred for 1 h at room temperature. After 3 h, microspheres were washed 3 times with ultrapure water and methanol, soaked in a methanol solution of 2-methylimidazole for 24 h, washed with methanol and ultrapure water, and dried to obtain EDTA–CS@ZIF-8 microspheres.

### 3.2. Adsorbent Characterization

Material characterization methods in this study were X-ray diffraction (XRD), Fourier transform infrared spectroscopy (FT-IR), scanning electron microscopy (SEM), X-ray photoelectron spectroscopy (XPS), and Brunaeur–Emmet–Teller (BET) surface area, pore volume, and pore-size analysis.

XRD was used to analyze the microscopic spatial crystal structure of solid materials. The type of crystal plane was determined by the obtained characteristic diffraction peaks, thereby determining the type of crystal structure and material. In this work, XRD analysis was performed on an X-ray diffractometer (XRD; D2 Phaser, Bruker, Yokohama, Japan) with Cu–Kα radiation, and scanning test range was set to 5°–50°. FT-IR (FTIR-4200, JASCO, Tokyo, Japan) was used to characterize the chemical structure and functional groups of the material. The KBr was ground and mixed with the dried sample, and the mixture was pressed into thin plates for infrared testing. The surface chemistry properties of EDTA–CS@ZIF-8 were investigated by X-ray photoelectron spectroscopy (XPS; K-Alpha, Thermo Scientific Center, Waltham, MA, USA). The surface morphology of the materials was observed by SEM (JCM-6000, JEOL Tokyo, Japan). The surface area was determined by the BET method. Samples were degassed at 423.15 K for 3 h, and then subjected to N_2_ adsorption/desorption experiments (N_2_-BET; TriStar 3020, Micromeritics, Norcross, GA, USA). Pore volume and size were calculated from the amount of adsorbed N_2_ at the relative pressure of 0.99. 

### 3.3. Adsorption Experiments

Aqueous solutions with different concentrations of La(NO_3_)_3_·6H_2_O, Eu(NO_3_)_3_·6H_2_O, and Yb(NO_3_)_3_·3H_2_O were used as simulated wastewater to study the adsorption characteristics of the EDTA–CS@ZIF-8 composite materials. These REE nitrate salts were diluted with ultrapure water to the required concentration. REE concentration in the filtrate was determined by inductively coupled plasma atomic emission spectrometry (ICP–AES; SPS1500, SEIKO, Chiba, Japan). Replicate experiments were performed 3 times. The operating conditions of ICP–AES are shown in Table 6. The absorption capacity of REEs for each sample was calculated by using Equation (10):(10)$$q_{e}=\left(C_{i}-C_{e}\right)\times \;V/m$$
where *q_e_* represents the adsorption capacity at equilibrium (mg g^−1^); *C_i_* and *C_e_* are the initial and equilibrium REE concentrations, respectively (mg·L^−1^); *V* is the volume of the solution (L); and *m* is the adsorbent mass (g) [42,43].

After the adsorption experiment, EDTA–CS@ZIF−8 (15 mg) was desorbed in 100 mL of a NaHCO_3_ (0.1 M) solution, and the regenerated adsorbent was used for cyclic adsorption experiments.

## 4. Conclusions

In this work, a new type of adsorption material, EDTA–CS@ZIF-8, was synthesized using chitosan, EDTA, and ZIF-8, and used to adsorb REEs from an aqueous solution. EDTA–CS@ZIF-8 was characterized by XRD, FT-IR, SEM, N_2_-BET, and XPS. The effects of various variables (including adsorbent dosage, temperature, pH value, adsorption time, and competitive ions) on REEs removal were evaluated. The results showed that the pseudo-second-order model and Langmuir isotherm could better describe the adsorption data of REEs on the adsorbent under the optimal adsorption conditions. The maximal adsorption capacity of EDTA–CS@ZIF-8 for REEs was estimated to be 256.4 mg·g^−1^ for La(III), 270.3 mg·g^−1^ for Eu(III), and 294.1 mg·g^−1^ for Yb(III), respectively. In addition, in the adsorption/desorption experiments, EDTA–CS@ZIF-8 exhibited high adsorption capacity for REEs, which proved that EDTA–CS@ZIF-8 could be an effective adsorbent for REEs. This leads to good application prospects in the removal of REEs from aqueous solutions. From the perspective of environmental protection, this brings very important information that can be used to treat industrial wastewater including pollutants, so it is a promising choice for the treatment of polluted water.

## Figures and Tables

**Figure 1 ijms-22-03447-f001:**
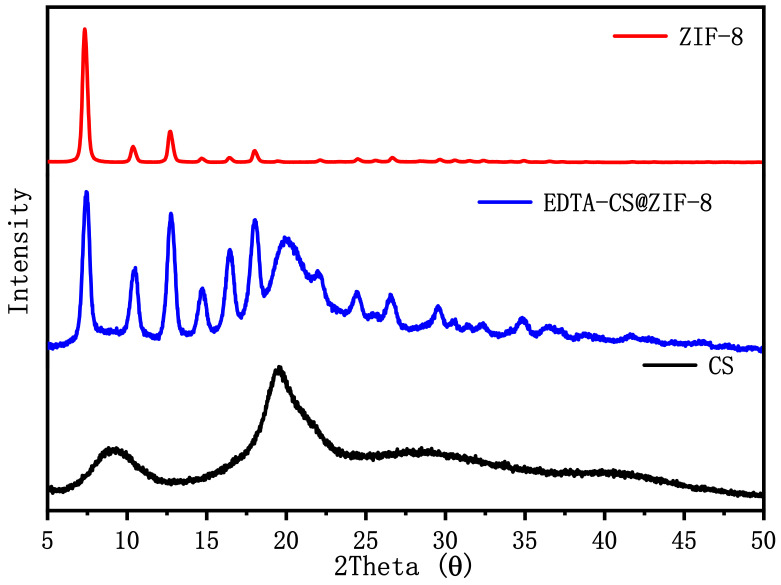
XRD of the zeolite imidazole framework (ZIF-8), chitosan, and ethylenediamine tetraacetic acid (EDTA)–chitosan (CS)@ZIF-8.

**Figure 2 ijms-22-03447-f002:**
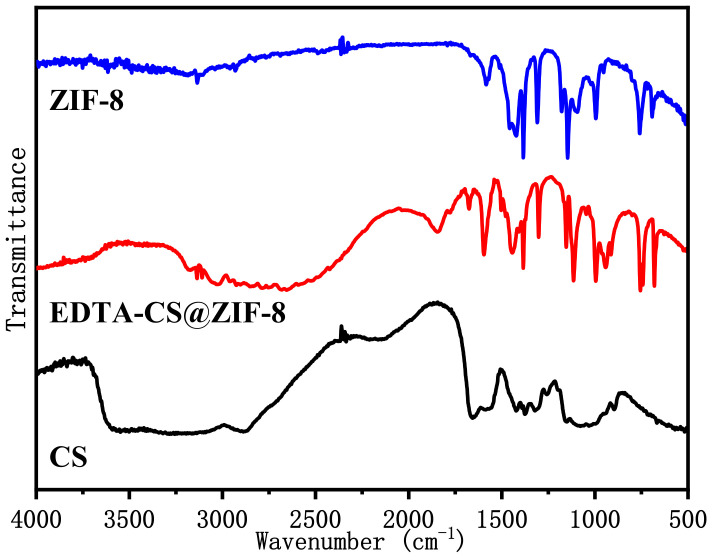
FT-IR of ZIF-8, chitosan, and EDTA–CS@ZIF-8.

**Figure 3 ijms-22-03447-f003:**
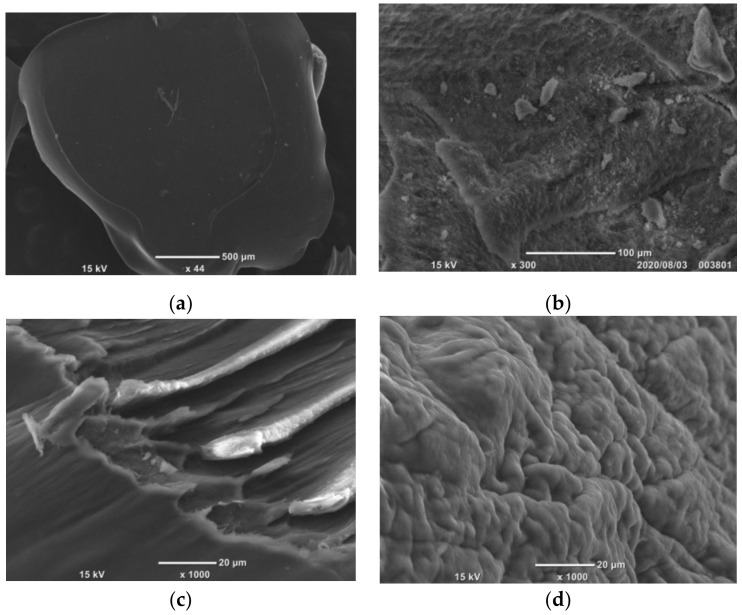
SEM images of (**a**,**c**) chitosan, (**b**) ZIF-8, and (**d**) EDTA–CS@ZIF-8.

**Figure 4 ijms-22-03447-f004:**
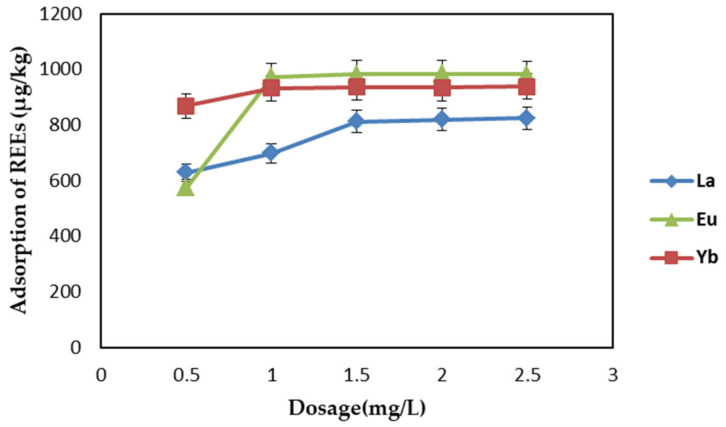
Effect of adsorbent dosage on adsorption of rare-earth elements (REEs) by EDTA–CS@ZIF-8.

**Figure 5 ijms-22-03447-f005:**
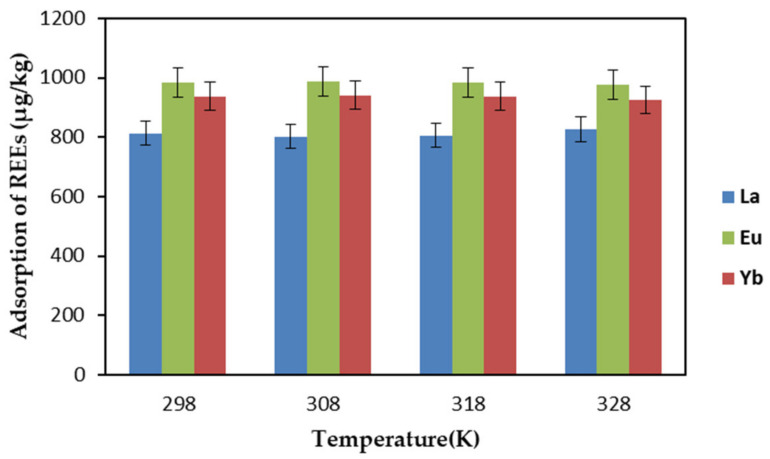
Effect of temperature on REEs adsorption of REEs by EDTA–CS@ZIF-8.

**Figure 6 ijms-22-03447-f006:**
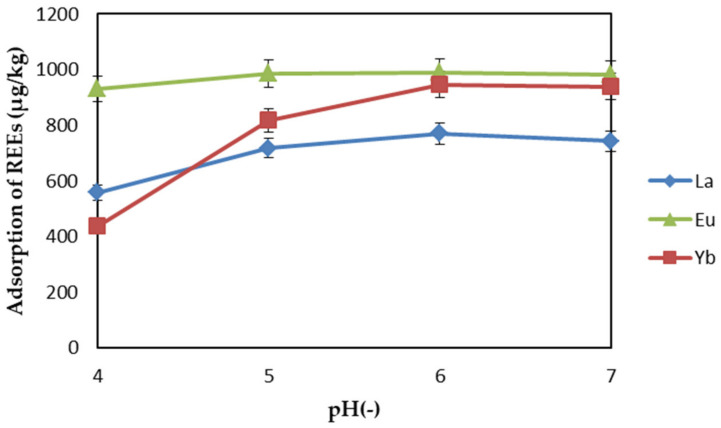
Effect of pH on REEs adsorption by EDTA–CS@ZIF-8.

**Figure 7 ijms-22-03447-f007:**
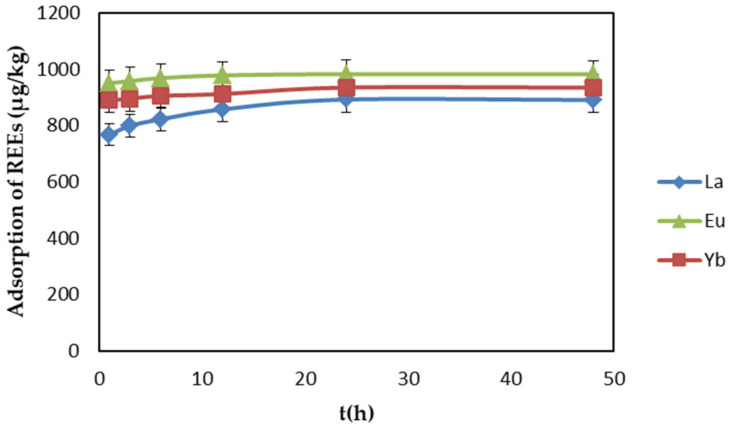
Effect of contact time on REEs adsorption by EDTA–CS@ZIF-8.

**Figure 8 ijms-22-03447-f008:**
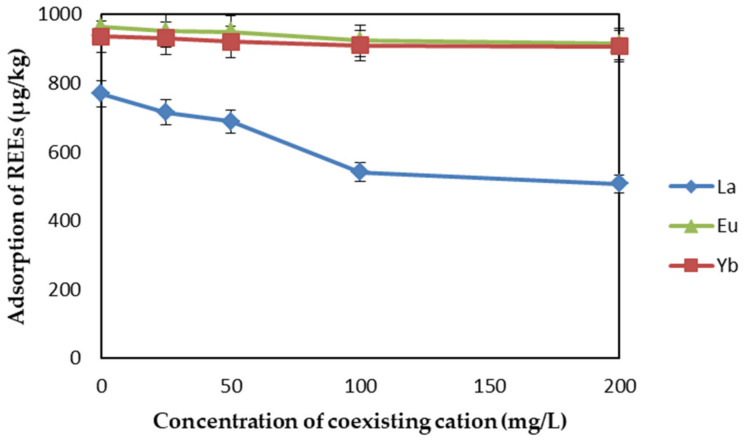
Effect of competitive ions (0, 25, 50, 100, and 200 ppm of Na^+^, K^+^, Ca^2+^, and Mg^2+^) on REEs adsorption by EDTA-CS@ZIF-8.

**Figure 9 ijms-22-03447-f009:**
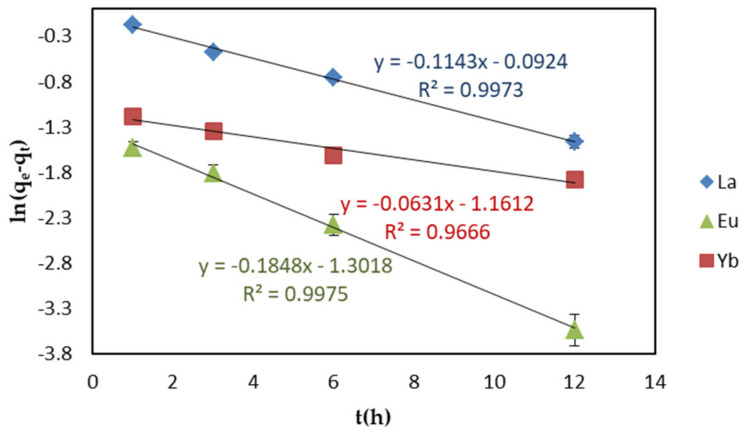
Pseudo−first−order linear kinetic models of REEs adsorption on EDTA–CS@ZIF-8.

**Figure 10 ijms-22-03447-f010:**
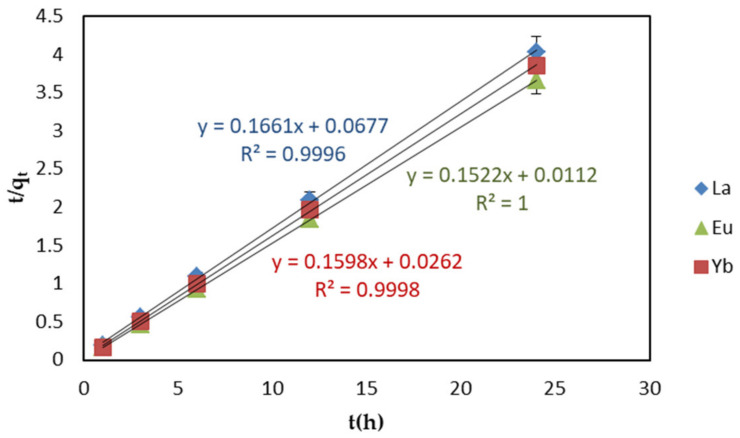
Pseudo−second−order linear kinetic models of REEs adsorption on EDTA–CS@ZIF-8.

**Figure 11 ijms-22-03447-f011:**
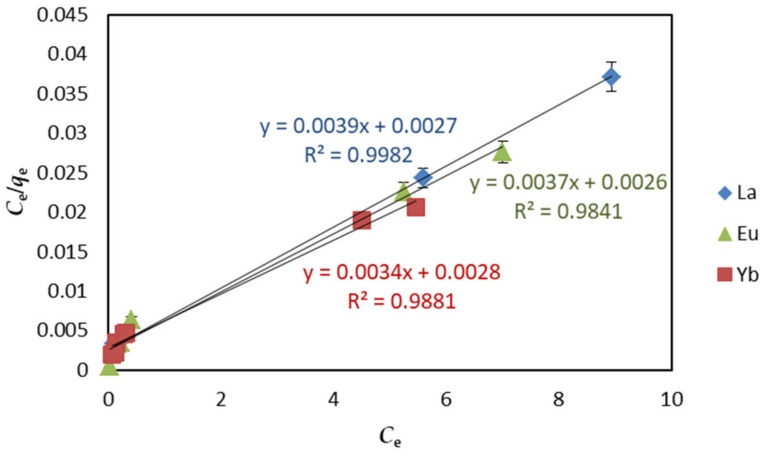
Langmuir isotherms of REEs adsorption on EDTA–CS@ZIF-8.

**Figure 12 ijms-22-03447-f012:**
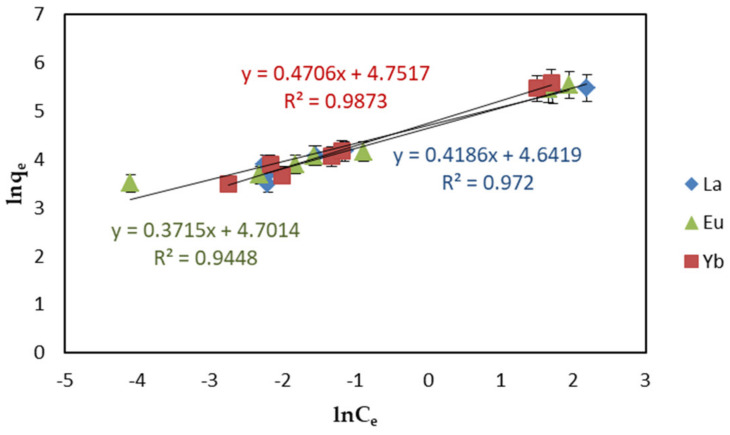
Freundlich isotherms of REEs adsorption on EDTA–CS@ZIF-8.

**Figure 13 ijms-22-03447-f013:**
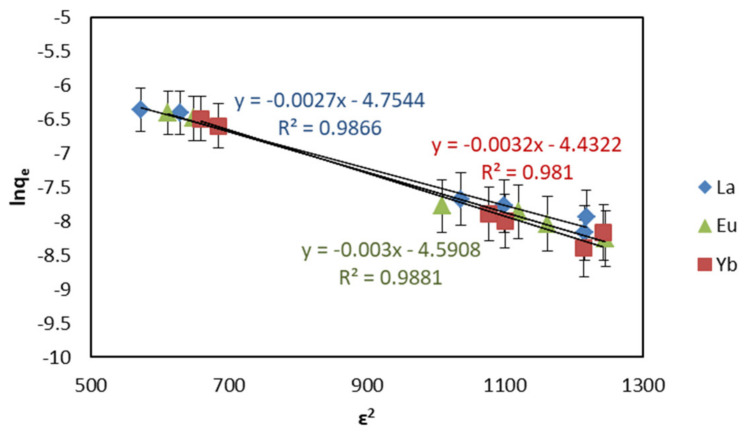
Dubinin–Radushkevich (D−R) isotherms of REEs adsorption on EDTA–CS@ZIF-8.

**Figure 14 ijms-22-03447-f014:**
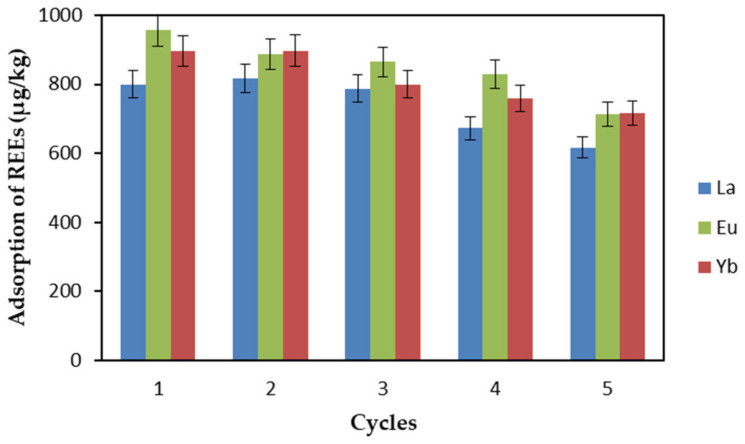
Recycled adsorption of REEs by EDTA-CS@ZIF-8.

**Figure 15 ijms-22-03447-f015:**
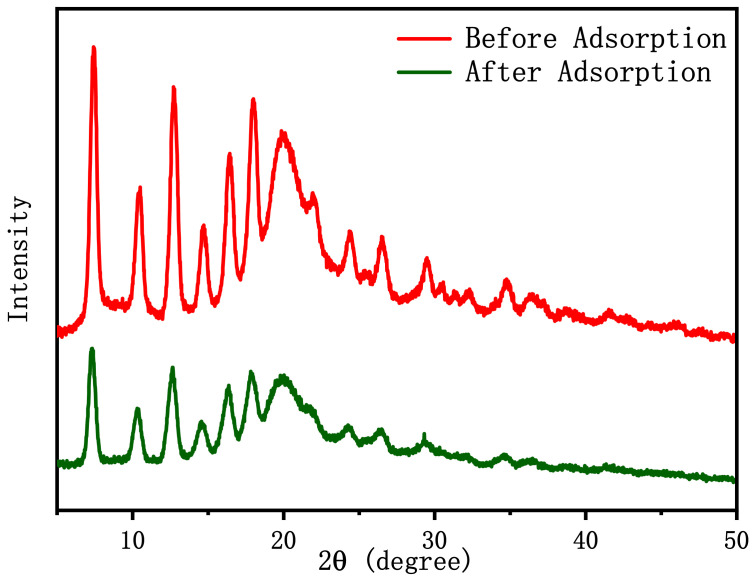
XRD of EDTA–CS@ZIF-8 composites before and after adsorption of Eu(III).

**Figure 16 ijms-22-03447-f016:**
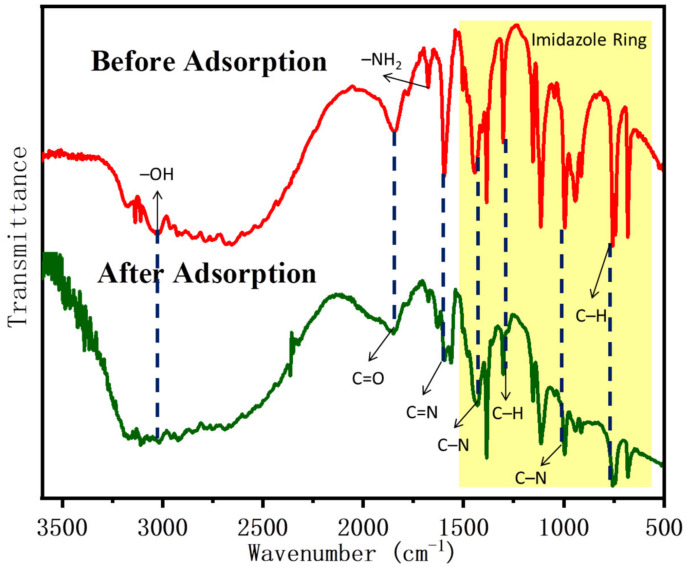
FTIR spectra of EDTA–CS@ZIF-8 composites before and after adsorption of Eu(III).

**Figure 17 ijms-22-03447-f017:**
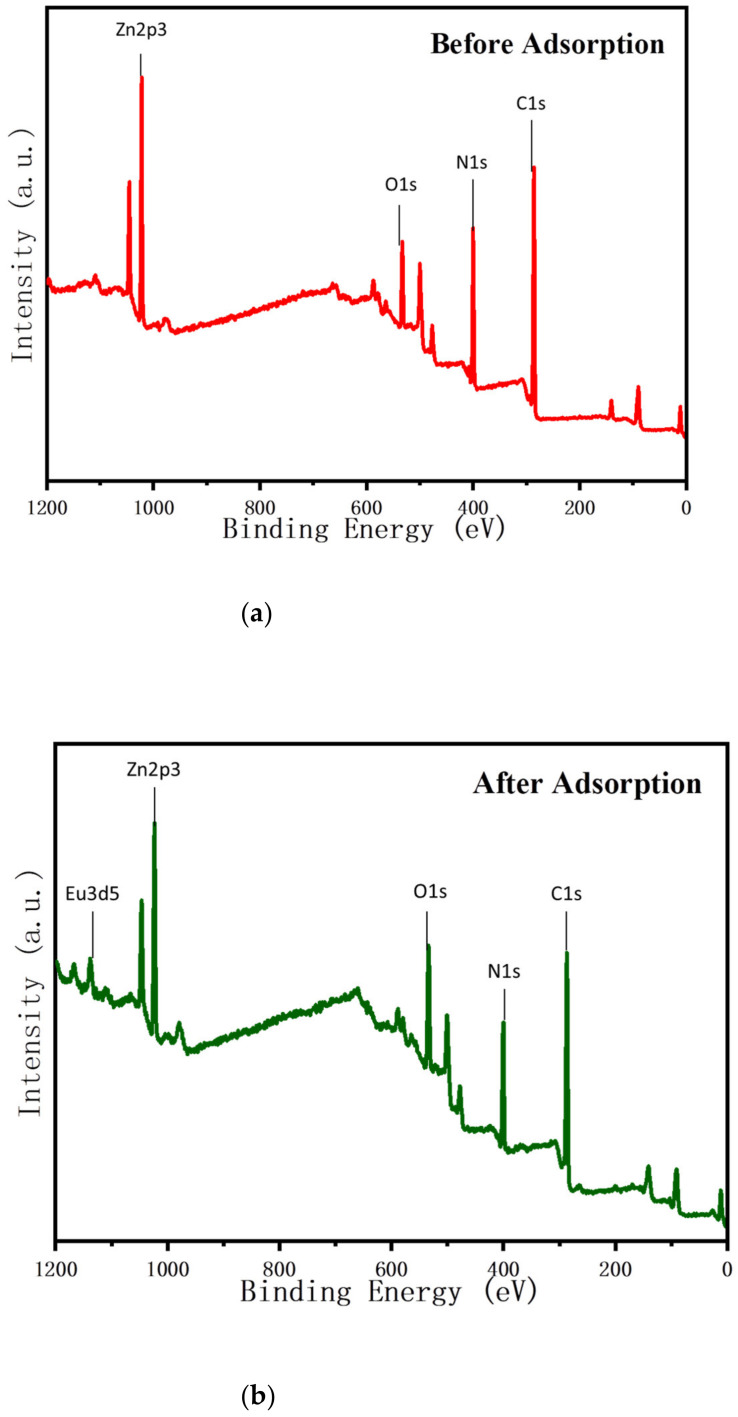
X-ray photoelectron spectroscopy (XPS) spectra of EDTA–CS@ZIF-8 before (**a**) and after (**b**) adsorption of Eu(III).

**Table 1 ijms-22-03447-t001:** Brunauer–Emmet–Teller (BET) surface area, pore size, and pore volume of ZIF-8, CS, and EDTA–CS@ZIF-8.

Sample	BET Surface Area (m^2^·g^−1^)	Pore Volume (cm^3^·g^−1^)	Pore Size (nm)
ZIF-8	1052	0.6360	2.416
CS	0.3429	0.001207	14.08
EDTA–CS@ZIF-8	29.314	0.01704	2.325

**Table 2 ijms-22-03447-t002:** Parameters for two kinetic models of adsorption of REEs by EDTA–CS@ZIF-8.

Scheme	*q*_exp_ (mg/g)	Pseudo-First-Order			Pseudo-Second-Order		
		*q*_e_ (mg·g^−1^)	*k*_1_ (h^−1^)	*R* ^2^	*q*_e_ (mg·g^−1^)	*k_2_* (g·mg^−1^·h^−1^)	*R* ^2^
**La(III)**	5.954	0.9117	0.1143	0.9973	5.920	0.4075	0.9996
Eu(III)	6.556	0.2720	0.1848	0.9975	6.570	2.068	1.000
Yb(III)	6.238	0.3131	0.06310	0.9666	6.258	0.9746	0.9998

**Table 3 ijms-22-03447-t003:** Coefficient of Langmuir, Freundlich, and Dubinin–Radushkevich isotherms for REEs by EDTA–CS@ZIF-8.

Sample	Langmuir Isotherm			Freundich Isotherm			Dubinin–Radushkevich Isotherm		
	*q*_max_ (mg·g^−1^)	*K_L_* (L·mg^−1^)	*R* ^2^	*K_F_* ((mg·g^−1^)·(dm^−3^·mg^−1^)^1/*n*^)	1/*n*	*R* ^2^	*q_0_* (mmol·g^−^^1^)	*E* (kJ·mol^−^^1^)	*R^2^*
La(III)	256.4	1.440	0.9982	103.7	0.4186	0.9720	8.614	13.61	0.9866
Eu(III)	270.3	1.420	0.9841	110.1	0.3715	0.9448	10.15	12.91	0.9881
Yb(III)	294.1	1.210	0.9881	115.8	0.4706	0.9873	11.89	12.50	0.9810

**Table 4 ijms-22-03447-t004:** Comparison of adsorption properties among several adsorbents in literatures.

Sample	Adsorbent	Final Concentration (mg·L^−1^)	*q*_max_ (mg·g^−1^)	Partition Coefficient (mg·g^−1^·mM^−1^)	Ref.
La(III)	Activated carbon	5.27 × 10^−2^	7.100 × 10^−2^	1.347	[45]
	Cr-MIL−101-PMIDA	115	37.4	0.3252	[46]
	Fe_3_O_4_/chitosan(Cys) NC	1	17	17	[47]
	Sargassum hemiphyllum	3.51 × 10^−2^	9.73 × 10^−2^	2.772	[48]
	ZIF-8 NPs	6.36	28.8	4.528	[49]
	EDTA-CS@ZIF-8	8.937	256.4	28.69	This study
Eu(III)	Activated carbon	0.0612	9.720 × 10^−2^	1.588	[45]
	Fe_3_O_4_@mSiO_2_-DODGA NPs	2	36.86	18.43	[50]
	Sargassum hemiphyllum	2.07 × 10^−2^	0.119	5.749	[48]
	SiO_2_/GA-g-PAM NC	2.5	10.11	4.004	[51]
	SiO_2_/PAA NC	15	268.8	17.92	[52]
	EDTA-CS@ZIF-8	7.007	270.3	38.53	This study
Yb(III)	Fe_3_O_4_/chitosan(Cys) NC	1	18.4	18.4	[47]
	Fe_3_O_4_@mSiO_2_-DODGA NPs	0.4	34.36	85.9	[50]
	Zr/XG-Zn-Al NC	8.25	25.73	3.119	[53]
	EDTA-CS@ZIF-8	5.459	294.1	53.87	This study

**Table 5 ijms-22-03447-t005:** Atomic ratio of EDTA–CS@ZIF-8 before and after adsorption of Eu(III) by XPS analysis.

Name Atomic%	Before Adsorption	After Adsorption
C1s	61.96	59.05
N1s	23.04	18.49
O1s	9.24	14.2
Zn2p3	5.75	5.34
Eu3d5	-	1.02

**Table 6 ijms-22-03447-t006:** Operating conditions of inductively coupled plasma atomic emission spectrometry (ICP-AES).

Parameters	Conditions
Rf frequency	27.12 MHz
Incident power	1.3 kW
Outer gas	17 dm^3^Ar min^−1^
Intermediate gas	0.55 dm^3^Ar min^−1^
Carrier gas	0.58 dm^3^Ar min^−1^
Observation height/mm	10.3 mm above work coil
Integration time	3 s
Detection wavelength/nm	379.48 (La(III)) 381.97 (Eu (III)) 328.94 (Yb (III))

## Data Availability

Data is contained within the article.

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
