# Peer review of "Adsorption of REEs from Aqueous Solution by EDTA-Chitosan Modified with Zeolite Imidazole Framework (ZIF-8)"

_ijms, 2021, doi:10.3390/ijms22073447_

Round 1

Reviewer 1 Report

This manuscript entitled "Adsorption of REEs from aqueous solution by EDTA-Chitosan modified with zeolite imidazole framework (ZIF-8)" by Feng et al has good concept. However some points to be reconsidered prior to make a final decision. 

  1. Introduction has been divided in several sections. It should be precise and focused only background information and necessity for the research conducted.
  2. No significance test has been conducted in any results. Include in every figure.
  3. Why the authors selected only Chitosan?
  4. TEM should be included in characterization studies.
  5. In Fig 1 & 2, font size including axis number and title is very small. Font size should increase.

Author Response

General Comments:

This manuscript entitled "Adsorption of REEs from aqueous solution by EDTA-Chitosan modified with zeolite imidazole framework (ZIF-8)" by Feng et al has good concept. However some points to be reconsidered prior to make a final decision.

 Response: Thank you for your comments. The comments are highly encouraging and helpful for our manuscript improvement. We are aware of the fact that there are still many shortcomings in our manuscript. We have checked the manuscript again and modified some parts which you pointed out.

Specific Comments:

  1. Introduction has been divided in several sections. It should be precise and focused only background information and necessity for the research conducted.

 Response: Thank you for your valuable comment. Based on your comment, we have modified the introduction to make the structure of the article more clearly by reducing some paragraphs.

2. No significance test has been conducted in any results. Include in every figure.

Response: Thanks for your valuable suggestion. We do agree with your comment. Based on your comment, significance test has been conducted in any results, and has been shown in each figure.

3. Why the authors selected only Chitosan?

Response: Thanks for your comment. We selected only chitosan because chitosan is a biomaterial, which has been a research hotspot in recent years. Chitosan is cheap and easy to obtain, which has been widely used to remove heavy metal ions in wastewater. However, chitosan has some disadvantages as follows: the mechanical strength of chitosan is not very high, so it can only be dissolved in acid medium, and the water solubility is also very poor, which limits the application of chitosan. Therefore, there is room to improve these material by modification in our work, then, we choose chitosan hoping to expand its application.

4. TEM should be included in characterization studies.

Response: Thanks for your valuable suggestion. However, we cannot conduct TEM analysis due to the lack of sample and the circumstances of the apparatus in this time.

In this work, we have conducted characterization by some methods (XRD, FT-IR, N2-BET including SEM analysis) and, moreover, our main purpose is to perform adsorption experiments of REEs and to investigate the adsorption mechanism rather than to perform characterization. TEM will be presented elsewhere in the future paper as needed.

5. In Fig 1 & 2, font size including axis number and title is very small. Font size should increase.

Response: Thanks for your comment. Based on your comment, we have modified the font of Figure 1 and 2.

Reviewer 2 Report

The article presents a interesting work removal of REEs using modified zeolite adsorben 

The results presented are significant and can be considered for publication

Author Response

Comment: The article presents an interesting work removal of REEs using modified zeolite adsorbent. The results presented are significant and can be considered for publication.

Response: Thank you very much for your comment. The comment is encouraging and helpful. We have checked the manuscript again and revised some parts.

Reviewer 3 Report

I have gone with the review and I have the following remarks:

  1. Additional isotherms models should be introduced: Langmuir-Freundlich model, Dubinin-Radushkevich model
  2. There is the lack of FT-IR spectra with sorbed lanthanides
  3. There is the lack of spectra XPS with sorbed lanthanides
  4. The author did not mention which competitive ions were studied

Author Response

Comments:

I have gone with the review and I have the following remarks:

Response: Thank you for your comments. The comments are highly encouraging and helpful for our manuscript improvement. We are aware of the fact that there are still many shortcomings in our manuscript. We have checked the manuscript again and modified some parts which you pointed out.

  1. Additional isotherms models should be introduced: Langmuir-Freundlich model, Dubinin-Radushkevich model.

 Response: Thank you for your valuable comment. We do agree with your comment. Based on your comment, we have also introduced Langmuir-Freundlich model and Dubinin-Radushkevich model (Pages 11-12 in revised manuscript). That is, adsorption behavior of lanthanides by EDTA-CS@ZIF-8 was also investigated by applying Langmuir-Freundlich and Dubinin-Radushkevich isotherm models to the data obtained. The results are shown in Figure 13 and Table 4 (Page 13 in revised manuscript).

2. There is the lack of FT-IR spectra with sorbed lanthanides.

Response: Thanks for your valuable suggestion. Based on your comment, FT-IR analysis of EDTA-CS@ZIF-8 has been conducted after adsorption of lanthanides as well as that before adsorption (Page 16 in revised manuscript) by taking europium (Eu) as an example.

3. There is the lack of spectra XPS with sorbed lanthanides.

Response: Thanks for your valuable suggestion. Based on your comment, XPS analysis of EDTA-CS@ZIF-8 has been conducted after adsorption of lanthanides as well as that before adsorption (Pages 16-17 in revised manuscript) by taking Eu(III) as an example.

4. The author did not mention which competitive ions were studied

Response: Thanks for your comment. We have already denoted that “In this study, the adsorption experiments of REEs were conducted under the presence of other competitive ions with different concentrations (i.e. 0, 25, 50, 100 and 200 ppm) of Na+, K+, Ca2+, Mg2+.” (Lines 259-261, Page 9 in original manuscript). However, we forgot to write kinds of competitive ions on the axis of abscissa. We have added the name of competitive ions downward the legend in this revised manuscript.

Round 2

Reviewer 1 Report

Accept in present form

Author Response

Comment: Accept in present form.

Response: Thank you very much for your comment. The comment is encouraging and helpful. We have checked the manuscript again and revised some parts.

Reviewer 3 Report

FTG IR spectra of the samples with lanthanides and empty are not compared, It is necessary to evaluate the particular bands, which differ!

Author Response

Comments: FT-IR spectra of the samples with lanthanides and empty are not compared. It is necessary to evaluate the particular bands, which differ!

Response: Thanks for your valuable comment. We have already denoted that “From FT-IR spectra in Figure 16, it is found that the vibration of –NH2 at 1,670 cm-1 almost disappeared after Eu(III) adsorption, and that the broad peaks at 1596 cm-1 and 3185 cm-1 are attributed to the stretching of –NH2 and –OH vibration. This indicates that there are a large number of –OH and –NH2 bonds on the surface of EDTA–CS@ZIF-8. These functional groups may have participated in the interaction with Eu(III) and have brought to the chelation of Eu(III) ions with imidazole and chitosan.” (Page 15 in the manuscript).

However, we forgot to write the particular bands in Figure 16. We have added the main particular bands in Figure 16 for comparing FT-IR spectra before and after adsorption of lanthanides in this re-revised manuscript. Moreover, the sentence in text (Page 15) is also slightly revised (The newly added parts have been highlighted in blue letter in our revised manuscript).